# Clinical Pharmacokinetics of Approved RNA Therapeutics

**DOI:** 10.3390/ijms24010746

**Published:** 2023-01-01

**Authors:** Seong Jun Jo, Soon Uk Chae, Chae Bin Lee, Soo Kyung Bae

**Affiliations:** Integrated Research Institute of Pharmaceutical Sciences, College of Pharmacy, The Catholic University of Korea, Bucheon 14662, Republic of Korea

**Keywords:** FDA-approved RNA therapeutics, absorption, distribution, metabolism, excretion

## Abstract

RNA-mediated drugs are a rapidly growing class of therapeutics. Over the last five years, the list of FDA-approved RNA therapeutics has expanded owing to their unique targets and prolonged pharmacological effects. Their absorption, distribution, metabolism, and excretion (ADME) have important clinical im-plications, but their pharmacokinetic properties have not been fully understood. Most RNA therapeutics have structural modifications to prevent rapid elimination from the plasma and are administered intravenously or subcutaneously, with some exceptions, for effective distribution to target organs. Distribution of drugs into tissues depends on the addition of a moiety that can be transported to the target and RNA therapeutics show a low volume of distribution because of their molecular size and negatively-charged backbone. Nucleases metabolize RNA therapeutics to a shortened chain, but their metabolic ratio is relatively low. Therefore, most RNA therapeutics are excreted in their intact form. This review covers not only ADME features but also clinical pharmacology data of the RNA therapeutics such as drug–drug interaction or population pharmacokinetic analyses. As the market of RNA therapeutics is expected to rapidly expand, comprehensive knowledge will contribute to interpreting and evaluating the pharmacological properties.

## 1. Introduction

In recent decades, RNA therapeutics have pioneered a broad range of medical uses in rare diseases related to gene expression as well as vaccines for combatting SARS-CoV-2 virus (COVID-19). The exclusive target specificity of RNA therapeutics can offer great advantages over traditional drugs by modulating the translation of disease-causing proteins via the downregulation of a gene target or by injecting synthetic mRNA to translate the encoded target protein [1,2]. Therefore, RNA therapeutics can be designed to target almost any genetic element, most of which may have previously been considered undruggable using other established drugs, including both small molecules and antibodies [3].

Over the last five years, numerous RNA therapeutics have been approved by the US Food and Drug Administration (FDA) for various applications (Figure 1). This review focuses on FDA-approved drugs that have oligonucleotide structures and can access pharmacokinetic/pharmacodynamic data as published information including FDA/EMA website database. Nusinersen, approved by the FDA in 2016, has proven efficacy in the treatment of spinal muscular atrophy and is the first approved drug for this disorder. Additionally, eteplirsen was the first approved drug to show activity in the treatment of some types of Duchenne muscular dystrophy (DMD). After the first drug approval for DMD, golodirsen, viltolarsen, and casimersen were approved by the FDA in 2019, 2020, and 2021, respectively, for other disorders (Table 1). Furthermore, after the first approval of patisiran—a small interfering RNA (siRNA) drug—in 2018, new siRNA drugs have been approved every year (Table 1).

RNA therapeutics differ from conventional drugs in terms of their pharmacological mechanisms of action and pharmacokinetic properties [4,5]. Understanding their absorption, distribution, metabolism, and excretion (ADME) is crucial for the development and safety assessment of new RNA therapeutics. Insight into these properties is important because most approved RNA therapeutics or candidates share similar modifications in their RNA structure and can exhibit similar ADME processes.

## 2. RNA Therapeutics Structure and Base Modification

RNA therapeutics are oligonucleotides consisting of RNA bases (adenine, cytosine, guanine, and uracil). Each base is linked to a phosphate backbone, which provides structural support to the strand. RNA therapeutics exhibit their therapeutic potency by binding to specific sites of pre- or mature RNA through Watson–Crick base pairing [6]. Based on their structure, RNA therapeutics can be divided into antisense oligonucleotides (ASOs), siRNA, mRNA, and aptamers. FDA-approved mRNA vaccines are not covered in this review because of the lack of traditional pharmacokinetic studies. Historically, the only FDA-approved aptamer-based drug was pegaptanib. Pegaptanib, a twenty-seven base RNA aptamer, inhibited vascular endothelial growth factor (VEGF) and was used to treat age-related macular degeneration. However, its use began to decline due to the development of more effective treatment options, such as ranibizumab, and eventually, and its marketing was discontinued [7].

Defibrotide, a polydisperse mixture of single- and double-stranded oligonucleotides with variable length and sequence composition, was the first approved RNA drug in 2016 and is derived from the porcine intestinal mucosa [8]. It was approved for the treatment of hepatic veno-occlusive disease and may protect the cells lining the blood vessels in the liver and prevent blood clotting. Defibrotide has multiple and complex mechanisms of action, with anti-inflammatory, anti-atherosclerotic, anti-ischemic, and antithrombotic properties [9].

Taken together, the majority of RNA therapeutics approved within the past five years have been ASOs or siRNA-based drugs. ASOs are short, single-stranded nucleic acids that are typically 15–30 bases in length [10]. siRNAs have a well-defined structure consisting of short (usually 20–24 base pairs) RNA duplexes with two base overhangs in the 3′ region [11].

Patisiran was the first approved siRNA drug in 2018 and formulated as a lipid-nanoparticle (LNP) for effective delivery to target. It consists of a 21-mer double-stranded siRNA (ALN-18328) encapsulated in four lipid excipients, including two novel lipid complex formulations: DLin-MC3-DMA and PEG_2000_-C-DMG. Owing to its limited chemical modifications, patisiran uses the LNP system for stability in circulation and effective liver uptake [12]. For sufficient encapsulation of LNP, the amount of DLin-MC3-DMA (6.76 mg) and PEG_2000_-C-DMG (0.76 mg) per 1 mg dose of ALN-18328 is considerable [13]. Therefore, the pharmacokinetic study of patisiran included both DLin-MC3-DMA and PEG_2000_-C-DMG in the amount of ALN-18328 [14].

Native oligonucleotides are rapidly degraded by nucleases and have a low affinity to proteins present in the plasma, resulting in their rapid elimination [15,16]. To overcome these issues, chemical modifications have been introduced into bases, RNA backbones, or sugar moieties (Figure 2). First, base modification is often applied to induce stronger Watson–Crick base pairing. This method also increases the thermal stability of duplexes formed between oligonucleotides and their target RNA [17]. However, higher affinity may increase the potential of off-target effect, contributing adverse effects. Additionally, modified pyrimidines have been shown to hinder siRNA-mediated silencing as they are relatively bulky, preventing proper attachment of the RNA-induced silencing complex (RISC) [18]. Second, the ribose modification in RNA is widespread method to improve stability against nuclease degradation, resulting in an increase in its pharmacokinetic half-life [19]. The most widely modified position in sugar moieties is their 2′-carbon. Common modifications include 2′-O-methyl (2′-O-Me), 2′-fluoro RNA (2′-F-RNA), and 2′-O-methoxyethyl (2′-O-MOE). Third, the phosphodiester linkages of the nucleic acid backbone are another target for modification because nucleases can cleave the linked phosphodiester bonds. The most common modification in the nucleic acid backbone is the phosphorothioate (PS) linkage, in which a non-bridging oxygen is substituted with sulfur. The advantage of the PS linkage is not only resistance against nuclease degradation, but also increased tissue uptake [20,21]. Another common modification for ASOs is the phosphorodiamidate morpholino oligomer (PMO), which contains a modified sugar of morpholine rings connected by phosphorodiamidate linkages [22]. PMO is resistant to various enzymes in biological fluids, enhancing the biological stability and efficacy of the RNA therapeutic [22].

## 3. Absorption and Administration Routes

RNA therapeutics are administered parenterally, commonly via the subcutaneous or intravenous route (Table 1 and Table 2). Bioavailability of RNA therapeutics after subcutaneous administration is low to intermediate compared with that of other drugs, which can be explained by nucleolytic degradation [23,24]. For example, systemic bioavailability of inclisiran after subcutaneous administration was determined to be 35.1–48.9% in rats and 24.7–33.8% in monkeys [25]. Following subcutaneous injection, these compounds are rapidly absorbed into the systemic circulation, with peak concentrations generally reached within 2–4 h (Table 2). Compared to intravenous injection, the plasma concentration following subcutaneous administration decreased more slowly after peaking owing to prolonged and continued absorption from the injection site [26]. In particular, subcutaneous givosiran linked with N-acetylgalactosamine (GalNAc) moieties is predominantly distributed to the liver, and the liver exposure ratio is significantly higher than that after intravenous administration, despite the drug’s low bioavailability in rats and monkeys [27].

Fomivirsen and pegaptanib are used to treat ocular diseases, which necessitates a local approach of administration to the drug target and optimization, rather than a systematic approach. Therefore, both drugs are administered via direct intravitreal injections. Based on a nonclinical study, the fomivirsen concentration of the retinal compartment was measured as 3.5 µmol/L and 0.88 µmol/L in the eyes of rabbits and monkeys, respectively, after single intravitreal administration [28,29]. Furthermore, pharmacokinetic modeling predicted that the peak and trough concentrations of the drug in the retina would range from 5 to 0.6 µmol/L following a bi-weekly administration of 165 µg to the human eye [30].

Systemically administered RNA therapeutics cannot cross the brain-blood barrier because of their bulky molecular size and negatively charged atoms in the backbone. To facilitate to effect on the central nervous system, nusinersen, used in the treatment of spinal muscular atrophy (SMA), is administered via intrathecal injection [31,32]. Administered nusinersen rapidly distributes in the cerebrospinal fluid (CSF) with a smaller volume of distribution (0.4 L) compared with its larger volume of distribution in plasma (29 L). The terminal half-life of nusinersen in the CSF was measured to be 3–6 months in monkeys and 4–6 months in children [33,34]. The infrequent maintenance dosing, which is once every 4–6 months, is supported by the long terminal half-life of the drug.

## 4. Distribution

RNA therapeutics interact with their RNA targets in the cytosol or nucleus of cells [35]. However, because of the hydrophobicity of the cell membrane, RNA therapeutics have a limited ability to diffuse from the blood into the peripheral tissues. Therefore, various cellular internalization pathways are necessary for RNA therapeutic uptake. Balanced plasma protein binding is required for effective cellular internalization. If RNA therapeutics bind plasma proteins too tightly, tissue distribution from the systemic circulation may be prevented [36]. In contrast, drugs that are less extensively bound to plasma proteins are cleared more rapidly, primarily due to metabolism in the blood or urinary excretion [37,38]. The introduction of a phosphorothioate backbone to RNA therapeutics increases their affinity to plasma proteins by ≥85% across all species, consequently increasing their half-life and promoting their uptake into systemic tissues [39]. Specifically, RNA therapeutics with PS linkages show high plasma protein binding: givosiran shows 90%, inclisiran shows 87%, inotersen shows 94%, lumasiran shows 85%, mipomersen shows 96%, and nusinersen shows 94% (Table 2). On the other hand, oligonucleotides that lack charge, such as PMOs, have relatively lower affinities (≤40%) for plasma proteins [38,40]. Unexpectedly, fomivirsen is only approximately 40% bound to monkey or rabbit vitreous proteins, despite its PS linkages [41].

For patisiran formulated as an LNP, plasma protein binding was assessed in two different ways. In the first method, the binding of patisiran to serum albumin and alpha-1-acid glycoprotein was measured to be 0.46% and 2.07%, respectively. In another method, human plasma protein binding of PEG2000-C-DMG alone, as an LNP PEG excipients, was determined to be ~97%. However, protein binding of patisiran followed the serum albumin and alpha-1-acid glycoprotein results because the molar ratio of PEG_2000_-C-DMG making up patisiran was much lower than that of other components [6].

The majority of RNA therapeutics are systemically distributed to the kidney and liver, which are primary excretion and metabolic organs [38,42,43]. As mentioned above, some RNA therapeutics that have GalNAc conjugates are primarily delivered to liver hepatocytes as their target tissue [44,45,46]. Additionally, RNA therapeutics that target muscle tissues, such as those used to treat DMD, are designed with high potency to overcome poor muscle tissue bioavailability [19,47,48].

## 5. Metabolism

Degradation of RNA therapeutics is typically mediated by exo- and endonucleases. Generally, unmodified oligonucleotides are rapidly degraded by 3-exonucleases [49,50,51]. As in the case of unmodified oligonucleotides, some RNA therapeutics are mainly metabolized by 3-exonucleolytic cleavage. The primary metabolite of givosiran, AS(N-1)3′ givosiran, is formed by 3′-end degradation of its antisense strand. Furthermore, it is an active metabolite that is detected in plasma in considerable amounts [27]. Therefore, a clinical pharmacokinetic study of givosiran measured the levels of AS(N-1)3′ givosiran exposure in plasma [52]. The concentrations of AS(N-1)3′ givosiran in plasma were approximately half the concentrations of givosiran. Other siRNA drugs, such as lumasiran and inclisiran, are also mainly metabolized to their respective AS(N-1)3′ metabolites [25,53]. However, the amount of each of their metabolites formed was much lower than that of givosiran. Nusinersen is also metabolized by a 3′-exonuclease, and its N-1 metabolites have been detected in both the CSF and plasma. In addition to 3′-exonucleases, 5′ exo- and endonuclease activities have also been observed in the metabolism of RNA therapeutics [54,55]. Moreover, mipomersen is degraded by endonucleases rather than by exonucleases. Several metabolites consistent with endonuclease-mediated cleavage have been detected in human urine samples collected from patients treated with mipomersen [38].

Of all the organs, drugs are largely distributed to the liver as this is the location of most drug metabolism. For this reason, some RNA therapeutics were tested using the s9 fraction of the liver, liver microsomes, or recombinant cytochrome P450 (CYP) systems to understand the metabolism of each drug. Most GalNAc siRNA drugs are not substrates of CYP enzymes; therefore, clinical interactions involving CYP enzymes are unlikely [56]. Additionally, several ASO drugs, including mipomersen, eteplirsen, nusinersen, and inotersen, do not undergo CYP-mediated metabolism.

In general, human radiolabeled ADME studies is a key study for new drug applications [57,58]. However, nonclinical studies of RNA therapeutics can be used as alternative approaches for conducting radiolabeled studies in humans. This is primarily driven by metabolites, which are fragments of the parent oligonucleotides formed by specific enzymes, such as exo- or endonucleases. This information can be obtained from studies in vitro research or animal mass balance studies. For eteplirsen, result of mouse radiolabeled study was compared with clinical distribution result without human radiolabeled study [59]. Similarly, metabolism of lumasiran was able to be investigated thoroughly using non-clinical data [53].

## 6. Elimination

Although renal clearance is the major excretion pathway for most RNA therapeutics, each individual RNA therapeutic has specific pharmacokinetic parameters related to their excretion (Table 2). For example, the rate of fomivirsen clearance from the vitreous humor is first-order with a half-life of approximately 55 h. The time-dependent decrease of fomivirsen levels in the vitreous humor may be due to its uptake into the retina and other ocular tissues or metabolism by nucleases [30]. Moreover, urinary excretion of mipomersen is minimal during the first 24 h after a single dose. Specifically, 1.38 to 3.30% is excreted by humans in the first 24 h following a 2 h intravenous infusion [60]. As the half-life of mipomersen is 31 days in humans, the elimination of mipomersen is delayed, indicating slow metabolism within tissues and subsequent excretion in urine [39,61]. Furthermore, inotersen is strongly bound to plasma proteins, and thus, glomerular filtration of inotersen is minimal. The renal excretion amount of intact inotersen within the first 24 h is less than 1% of the administered dose, whereas that of chain-shortened inotersen accounts for 13.5% of the administered dose [62]. As the elimination half-life of inotersen is 32.3 days, it appears that the metabolism or excretion of inotersen takes longer after the early distribution phase. Nusinersen is also excreted via the renal excretion pathway, and its metabolite is a nucleolytic product [63]. As mentioned above, nusinersen is administered intrathecally and must move to the blood for renal excretion. However, the elimination half-life of nusinersen can be prolonged due to the blood-CSF barrier from the CSF to the blood. All PMO RNA therapeutics approved by the FDA, including casimersen, eteplirsen, golodirsen, and viltolarsen, are excreted rapidly in the urine as unchanged drugs. The percentage of each drug recovered in the urine over 24 h was over 60%. The elimination half-life and plasma clearance are 3.5 h and 180 mL/h/kg for casimersen, 1.6–3.6 h and 200–300 mL/h/kg for eteplirsen, 3.5 h and 338–405 mL/h/kg for golodirsen, and 2.5 h and 217 mL/h/kg for viltolarsen, respectively [59,64,65,66].

GalNAc-conjugated siRNA drugs, such as givosiran, inclisiran, lumasiran, and vutrisiran, have been approved by the ASO and Drug Administration. Based on non-clinical data, the initial blood clearance of siRNA drugs appears to occur through high distribution to the liver. Givosiran is cleared from the plasma by conversion to AS(N-1)3′ givosiran (36%), uptake in the liver (52%), and urinary excretion (12%) [67]. It has been estimated that approximately 82.5% of elimination of inclisiran from plasma is due to hepatic uptake [68]. Additionally, the majority of lumasiran is also taken up by the liver and only 17.4% to 25.8% of the administered dose is excreted in the urine as unchanged lumasiran [69]. Urinary excretion of vutrisiran is less than 25% of the total dose administered [70]. However, the elimination half-life of GalNAc-conjugated siRNA drugs in the liver is estimated as 82.5 days for inclisiran (monkey data) and 66.9 days for lumasiran (PBPK modeling prediction). These slow elimination rates suggests that the fecal excretion of GalNAc-conjugated siRNA drugs was a small fraction and the major excretion pathway of these drugs could be renal excretion. To investigate whether kidney function may lead to changes in pharmacokinetic (PK) profiles, renal excretion, or pharmacodynamic (PD) outcomes of GalNAc-conjugated siRNA, a 5/6 nephrectomized rat study was conducted [71,72]. Experiments on the 5/6 nephrectomy rat model, representing moderate to severe renal impairment, demonstrated that the PK profile in the liver or observed PD outcomes were significantly unaffected by a reduction in urinary excretion [73]. Therefore, renal impairment is unlikely to influence the liver PK profile and subsequent PD outcomes of GalNAc-conjugated siRNA. The similar results were found in clinical research on inclisiran in patients with renal impairment [74]. For givosiran and lumasiran, the result of subgroup analyses showed mild to moderate renal impairment did not seem to influence efficacy [60,67].

## 7. Drug-Drug Interactions

Most RNA therapeutics have not been reported to be inhibitors or inducers of the CYP system yet, which causes classical drug–drug interactions. Therefore, they are not predicted to interact with small-molecule drugs primarily cleared through oxidative metabolic pathways. However, a clinical study on an indirect drug–drug interaction of givosiran has been reported [75]. In this study, givosiran was able to potentially affect the hepatic heme content, a prosthetic moiety of CYP enzymes and a cofactor for enzyme function, due to its pharmacodynamic effect. The results showed that givosiran had little to no impact on the activity of CYP2C9, weakly reduced the activity of CYP3A4 and CYP2C19, and moderately reduced the activity of CYP2D6 and CYP1A2. The interesting result is that an in vitro study using human liver microsomes indicated that givosiran did not directly inhibit any of the major CYP isoforms because it is difficult to demonstrate pharmacodynamic properties in vitro [56].

Transporters can also have clinically relevant effects on the pharmacokinetics and pharmacodynamics of a drug by modulating its absorption, distribution, and elimination [76]. In contrast to drug-metabolizing enzymes that are primarily expressed in the liver and small intestines, transporters are expressed throughout the human body and control the movement of substances to various tissues, including the gastrointestinal tract, liver, kidney, and brain [77]. Therefore, the drug–drug interaction potential mediated transporters have been evaluated for some RNA therapeutics during their development. Most GalNAc siRNAs were determined not to be substrates of clinically relevant transporters and demonstrated no inhibitory interactions with either uptake (OATP1B1, OATP1B3, OAT3, OCT1, and OCT2) or efflux (BCRP, BSEP, MATE1, MATE2-K, and P-gp) transporters, except for the concentration-dependent inhibition of P-gp by givosiran [57]. However, the administration route for givosiran is subcutaneous, and the value of I/IC50 is lower than the cutoff (0.03) proposed by Ellens and Lee [78,79]. Thus, the potential for givosiran to inhibit P-gp is not clinically significant. ASO drugs, including casimersen, eteplirsen, golodirsen, inotersen, mipomersen, nusinersen, and viltolarsen, have been studied in vitro in transporter-mediated pharmacokinetic drug interactions [59,64,65,66,80,81,82]. The results showed that ASO drugs are not substrates or inhibitors of a variety of human transporters and are unlikely to interact with other drugs due to competition or inhibition.

Some RNA therapeutics are highly bound to plasma proteins, and thus, protein-binding displacement can be considered another potential cause of pharmacokinetic interactions [83]. However, RNA therapeutics have some factors that were not influenced by protein binding displacement. First, RNA therapeutics, given that the dosing frequency is monthly or longer, are in circulation only transiently (a few hours) before being distributed into tissues, so the effect of plasma protein binding is minimal [17]. Second, clinical peak concentrations of RNA therapeutics are well below those of plasma proteins, such as albumin (~600 µM), that bind to most drugs [84,85]. Third, the binding of oligonucleotides to plasma proteins is relatively weak, and the binding sites for these hydrophilic drugs can differ from those of small molecular hydrophobic drugs [86,87]. Consequently, plasma protein binding has little to no impact on drug–drug interactions with RNA therapeutics.

## 8. Pharmacokinetic-Pharmacodynamic (PK/PD) Relationships

PK/PD analyses link dose-concentration relationships (pharmacokinetics, PK) and concentration-effect relationships (pharmacodynamics, PD), thereby predicting the effect of drug dosing over time [88]. Biomarkers can be used to examine the link between drug regimens and target effects for accurate PD analysis. As RNA therapeutics are designed to interact with specific target genes, their primary biomarkers are relatively clear. For example, inotersen was designed to be highly specific for transthyretin (TTR) mRNA, and the PD biomarker is the change in plasma TTR protein [89]. Similarly, inclisiran is chemically synthesized against the PCSK9 protein related to hyperlipidemia, and the PD biomarker is the concentration of the PCSK9 protein [25]. In addition, deletion mutations that cause DMD disrupt the dystrophin mRNA-reading frame and prevent the production of normal proteins. Therefore, the PD biomarker of ASO drugs used in the treatment of patients with DMD is the level of dystrophin production.

Although the PD biomarkers for RNA therapeutics are well described, the PK/PD relationship is more complex than that of conventional small molecules The complex PK/PD relationship is also shown in biologics and target-mediated drug disposition model was presented to explain this relationship [90]. Following the biologics, the PK/PD model for RNA therapeutics using target-mediated drug disposition model was developed recently [91]. The unusual property of PK/PD relationship is that plasma PK does not correlate directly with the PD effect [73]. First, the plasma PK T_max_ was significantly different from the corresponding PD T_max_. For example, the plasma PK T_max_ of vutrisiran was 0.2–12 h in healthy volunteers, whereas nadir TTR levels, considered as PD T_max_, were achieved by 50–90 days [70]. Second, there is a gap between the short plasma half-life and the prolonged duration of PD remaining for weeks or several months. As mentioned above, ASO drugs for DMD have a short half-life (<4 h), but their clinical effect was established by intravenous administration once weekly [92]. This means that the effect of the drugs remains sufficient despite their rapid elimination from the plasma.

It is necessary to determine other PK profiles to resolve this discordance in the plasma profile with PD activity. It was found that PD activity is better correlated with drug concentration in organs than plasma concentration if the RNA therapeutics had target organs [93]. For siRNA drugs, PD effect profiles correspond significantly more directly with RISC concentration in nonclinical studies because RISC is an essential structure for its function [73]. Although the concentration-effect relationship has not been thoroughly elucidated, some PK/PD models for RNA therapeutics have been developed and are covered in the next section.

## 9. Population Pharmacokinetic Analysis

Population pharmacokinetic or pharmacokinetic-pharmacodynamic (PK/PD) modeling and simulation has been applied in the clinical evaluation of some RNA therapeutics approved over the last decade. DMD treatment therapeutics had been approved using limited Pop PK or without PK/PD model because DMD is an ultra-rare disease, and most patients are too young to get sample sufficient for effective analysis [59,65,66]. This method supports the integration of studies with rich or sparsely sampled pharmacokinetic data and can be used to develop models; explain individual variability, such as demographic values, hepatic or renal function, and genotype; and guide the selection of appropriate dosing regimens [94]. Therefore, population PK is useful for developing RNA therapeutics to treat rare diseases related to genetic defects. Such patients are not only difficult to recruit as clinical subjects, but also vary in age, including children.

The two published population PK/PD studies were for inotersen and patisiran, which were approved for the same disease. A population PK/PD model for inotersen was developed by on data from phase I, II, and III clinical trials [95]. The PK/PD relationship for inotersen has been well-established compared to other RNA therapeutics because the half-life of inotersen is sufficiently long to link the plasma PK profile and TTR level using a two-compartment model [95]. Population PK analyses have shown that only body weight or body size is a significant covariate of inotersen clearance and volume of distribution, but does not present clinically relevant effects on inotersen exposure [95]. The analysis also showed that renal function was not a statistically significant covariate of inotersen clearance in patients with mild-to-moderate renal impairment [95]. In addition, the PK/PD model simulated four dosing regimens: 300 mg weekly with or without loading dose, 300 mg bi-weekly, and 150 mg weekly. The simulation suggests that the two lower doses would be effective, and the loading dose may not be necessary for chronic treatment with inotersen because it has little effect after 3 months [95].

The first population PK/PD model of an siRNA therapeutic was developed for patisiran [96]. The PK/PD relationship for patisiran was also established because the model used plasma concentrations linked to the RISC amount in hepatocytes. In clinical studies of patisiran, a 7–14 day lag period was observed between the peak pharmacodynamic effect and peak plasma concentrations. The PK/PD model for patisiran used a separate compartment to describe this observed hysteresis, and the resulting model adequately described the TTR time course following various dose regimens [96]. Model simulations indicated that patisiran showed similar pharmacological activity, regardless of genotype status of the patient [96]. The PK/PD model supports the currently approved patisiran dosing regimen of 0.3 mg/kg every three weeks (Q3W) for patients weighing less than 100 kg, because the simulation showed no effect of the two-fold higher PK exposures on the pharmacodynamic result over the body weight range 36.2–110 kg [96]. Furthermore, similar pharmacodynamic results were predicted with a capped fixed dose of 30 mg Q3W and a body weight-based dose of 0.3 mg/kg Q3W in patients weighing more than 100 kg, suggesting that 30 mg as the maximum dose is appropriate for patients weighing 100 kg or more.

## 10. The Drawbacks of RNA Therapeutics

The drawbacks of RNA therapeutics were related to their stability, rapid degradation, and delivery to target cells [97]. As mentioned earlier, the modification in RNA backbone structure allowed to improve the stability and prevent rapid degradation. In addition, various delivery systems using lipid-based or polymer-based system were introduced to overcome the lipid bilayer as well as to protect from degradation [98,99]. As a result, siRNA drugs using LNP-based delivery system and GalNAc-conjugated moiety drugs were approved by FDA. This means applicated delivery system in approved drugs can carry RNA molecules to target cells efficiently. However, the immunogenicity of LNP-based delivery system is considered as another safety concern [100,101]. Polyethylene glycol as main component of LNP induce additional antibodies, and the generation of antibodies that specifically bind to PEG results in reduced treatment efficacy or adverse drug reactions [102]. Antibodies against PEG after patisiran administration were also observed in the clinical results of patisiran, but the response was transient [103]. Although, reduced therapeutic efficacy or severe immune response related with generated antibodies were not observed in clinical study, the immunogenicity can result in toxicity. On the other hand, enhancing immunogenicity by LNP-based delivery system can be an advantage for mRNA vaccines, which need efficient antigen presentation [104]. The adjuvanticity of LNP-based delivery system can be one of the reasons for use in mRNA COVID-19 vaccines. After first approval of siRNA using LNP-based delivery system, other approved siRNAs use GalNAc-conjugated system rather than LNP-based delivery system, which seems because of the immunogenicity. However, further research about conjugates for various target is needed as no other conjugates for siRNA delivery have been approved.

## 11. Conclusions

The development of RNA therapeutics has advanced significantly over the last decade, resulting in the recent approval of several new drugs by the FDA and/or European Medicines Agency (EMA). Furthermore, numerous new RNA therapeutics are in clinical trials and are expected to be approved in the next decade, owing to being able to address unmet clinical needs about rare disease. Hence, a comprehensive understanding of the clinical pharmacokinetic properties of these drugs is needed to evaluate their therapeutic efficacy and safety. In particular, RNA therapeutics have distinctive ADME properties that can be distinguished from those of small-molecule drugs. Parenteral administration and a long terminal half-life are the most pronounced of these pharmacokinetic distinctions, in addition to their pharmacological specificity. Prolonged half-life caused by base modification can contribute to convenient dosing with regimens as infrequent as once a week to twice a year. Drug-drug interactions are rare because the metabolism of RNA therapeutics is not related to traditional drug-metabolizing enzymes or transporters. Instead, interactions caused by the manipulation of gene expression may continue to be relevant after treatment with RNA therapeutics. Several challenges regarding the interpretation of the PK/PD relationship and its application to various models remain, but some models have been successfully developed during drug development. Although, diverse drawbacks of RNA therapeutics can be solved by chemical modification and development delivery system, several advances are still needed. Further knowledge of the pharmacokinetic mechanism would support dose selection, dosing interval and a suitable formulation selection. Ultimately, new drug development for the clinical use of this class of drugs can be improved.

## Figures and Tables

**Figure 1 ijms-24-00746-f001:**
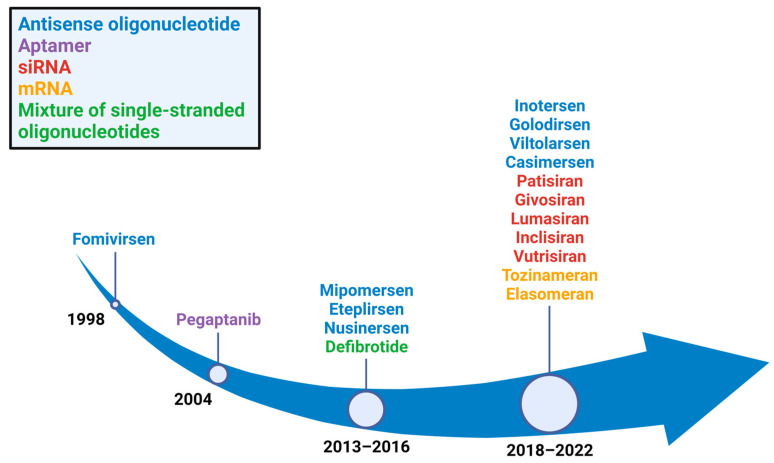
Years in which RNA therapeutics gained FDA approval. Eight antisense oligonucleotide drugs (blue), five single interfering RNA drugs (red), two mRNA drugs (orange), one aptamer drug (purple), and a mixture of single-stranded oligonucleotide drugs (green) were approved.

**Figure 2 ijms-24-00746-f002:**
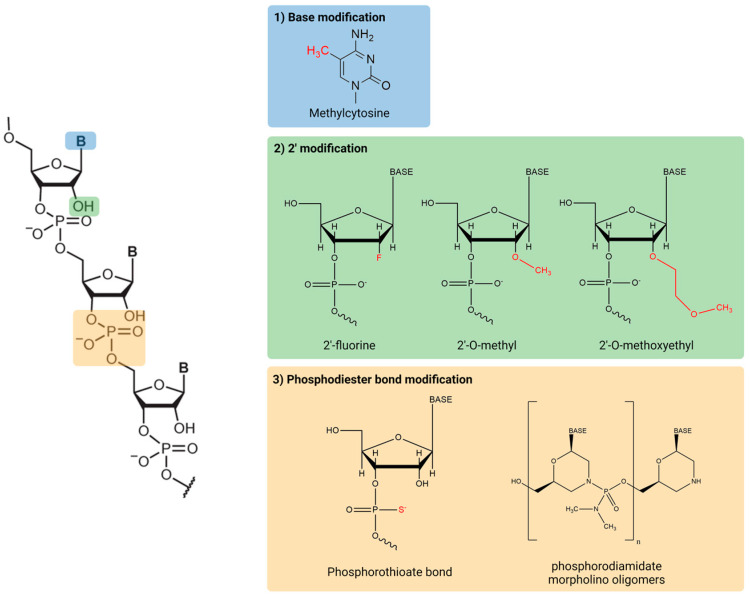
Common chemical structures of RNA therapeutics.

**Table 1 ijms-24-00746-t001:** RNA therapeutics that have been approved for clinical use.

Type	Drug	FDA Approval	Indication	Administration Route	Chemistry
ASO	Fomivirsen	1998	Cytomegalovirus retinitis in immunocompromised patients	Intravitreal	Phosphorothioate linkage
	Mipomersen	2013	Homozygous familial hypercholesterolemia	SC	2′-O-methoxyethylribonucleotides Deoxyribonucleotides Phosphorothioate linkage
	Nusinersen	2016	Spinal muscular atrophy	Intrathecal	2′-O-methoxyethylribonucleotides Phosphorothioate linkage
	Eteplirsen	2016	Duchenne muscular dystrophy	IV	Phosphorodiamidate morpholino oligomer
	Defibrotide	2016	Veno-occlusive disease	IV	Mixture of single and double stranded oligonucleotides
	Inotersen	2018	Hereditary transthyretin-mediated amyloidosis	SC	2′-O-methoxyethylribonucleotides Deoxyribonucleotides Phosphorothioate linkage
	Golodirsen	2019	Duchenne muscular dystrophy	IV	Phosphorodiamidate morpholino oligomer
	Viltolarsen	2020	Duchenne muscular dystrophy	IV	Phosphorodiamidate morpholino oligomer
	Casimersen	2021	Duchenne muscular dystrophy	IV	Phosphorodiamidate morpholino oligomer
siRNA	Patisiran	2018	Hereditary transthyretin-mediated amyloidosis	IV	Deoxyribonucleotides 2′-OMe ribonucleotides Lipid nanoparticle
	Givosiran	2019	Acute hepatic porphyria	SC	2′-F ribonucleotides 2′-OMe ribonucleotides Phosphorothioate linkage GalNAc moiety
	Lumasiran	2020	Primary hyperoxaluria type 1	SC	2′-F ribonucleotides 2′-OMe ribonucleotides Phosphorothioate linkage GalNAc moiety
	Inclisiran	2021	Primary hypercholesterolemia	SC	2′-F ribonucleotides 2′-OMe ribonucleotides Phosphorothioate linkage GalNAc moiety
	Vutrisiran	2022	Hereditary transthyretin-mediated amyloidosis	SC	2′-F ribonucleotides 2′-OMe ribonucleotides Phosphorothioate linkage GalNAc moiety

**Table 2 ijms-24-00746-t002:** Pharmacokinetic parameters of the RNA therapeutics.

Type	Drug	T_max_ (h) ^a^	Volume of Distribution	Protein Binding (%)	t_1/2_
ASO	Fomivirsen	-	-	40 (vitreous)	55 h
	Mipomersen	-	48.3 L/kg	>85	30 d
	Nusinersen	1–6	CSF: 0.4 L Plasma: 29 L	CSF: <25 Plasma: >94	CSF: 133–177 d Plasma: 68–87 d
	Eteplirsen	-	0.601 L/kg	6.1–16.5	1.6–3.6 h
	Defibrotide	-	8.1–9.1 L	93	2 h
	Inotersen	2–4	293 L	94	32.3 d
	Golodirsen	-	0.668 L/kg	33–39	3.5 h
	Viltolarsen	-	0.300 L/kg	39–40	2.5 h
	Casimersen	-	0.367 L/kg	8.4–31.6	3.5 h
siRNA	Patisiran	-	0.26 L/kg	2.1	3.2 d
	Givosiran	3 (0.5–8)	10.4 L	90	6 h
	Lumasiran	4 (0.5–12)	4.9 L	85	5.2 h
	Inclisiran	4	500 L	87	9 h
	Vutrisiran	4	10.1 L	80	5.2 h

^a^ T_max_ is applicable for subcutaneous administration. Data were extracted from EMA assessment reports or FDA approval packages for each drug.

## Data Availability

Not applicable.

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
