# Peer review of "Clinical Pharmacokinetics of Approved RNA Therapeutics"

_ijms, 2023, doi:10.3390/ijms24010746_

Round 1
Reviewer 1 Report
The authors have done a very good job is compiling literature on RNA therapeutics and pulling it together comprehensively. The paper is written well, is interesting and relevant.
I have some suggestions -
Compare RNA therapeutics to other biologics - perhaps monoclonal antibodies. Comparing to small molecules is less relevant. Or compare to the chemical drugs and other biologics.
Stay away from giving advice on how to make RNA drugs unless you are confident and have experience with this
Please read the literature you are quoting - the FDA guidance is misquoted/misunderstood.
I have made detailed suggestions/comments in the text - please consider making those changes.
Good luck!

Author Response
We are greatly encouraged by your assessment that our manuscript was interesting and relevant. Thank you for your thoughtful review and penetrating comments. We sincerely appreciate your effort and time for reviewing our manuscript. We will upload the author response as a Word file.

Reviewer 2 Report
Thank you for giving me this chance to review this interesting paper. The paper can be considered for publication if the following comments are addressed:
- Figure 1: arrows and use of dates beside drug name and in down in the main year arrow is a bit confusing to me.
- It was not clear to me how the authors searched for these drugs and what criteria did they use to include them. Therefore, the author should mentioned there search strategies to ensure all related paper were included.
- The authors need to recheck there writing to avoid plagiarism as some sentences and ideas was clearly plagiarized.
Author Response
We thank the reviewer for the positive assessment as well as for you time and efforts in reviewing and improving this manuscript. We have carefully considered the comments and concerns made by the reviewer. We will upload the author response document as a Word file.
Reviewer 3 Report
The current manuscript addresses the pharmacokinetics of RNA therapeutics, highlighting their potential by focusing on already marketed medications. Nevertheless, despite many advantages, these medications have their drawbacks, especially in what concerns their incorporation into a formulation. Hence, please discuss the limitations in formulating RNA, comparing it to small molecular weight drugs. Also, include a paragraph indicating the most suitable formulations for RNA, and why: are they nanoparticles, or other type of formulations? What kind of excipients are used in marketed formulations?
Author Response
Reviewer comment 3
Comments and Suggestions for Authors
The current manuscript addresses the pharmacokinetics of RNA therapeutics, highlighting their potential by focusing on already marketed medications. Nevertheless, despite many advantages, these medications have their drawbacks, especially in what concerns their incorporation into a formulation. Hence, please discuss the limitations in formulating RNA, comparing it to small molecular weight drugs. Also, include a paragraph indicating the most suitable formulations for RNA, and why: are they nanoparticles, or other type of formulations? What kind of excipients are used in marketed formulations?
Response: Thank you very much for suggesting new perspective about our topic. Your comments and suggestions are valuable and very helpful for revising and improving our manuscript. In response to your comments, we have added the section of the drawbacks of RNA therapeutics in revised manuscript. In the section, not only the limitation of RNA therapeutics but also introduced formulation techniques to resolve the drawbacks have been covered. Again, we are grateful for all of the insightful concerns and your help with making this a better manuscript.